# Microstructure and Wear Behavior of Laser Cladded Ni45 + High-Carbon Ferrochrome Composite Coatings

**DOI:** 10.3390/ma13071611

**Published:** 2020-04-01

**Authors:** Jiayang Gu, Ruifeng Li, Shungao Chen, Yuhao Zhang, Shujin Chen, Heng Gu

**Affiliations:** 1Marine Equipment and Technology Institute, Jiangsu University of Science and Technology, Zhenjiang 212003, China; gujiayang@126.com; 2School of Materials Science and Engineering, Jiangsu University of Science and Technology, Zhenjiang 212003, China; shungaoc@163.com (S.C.); yuhaozhang99@126.com (Y.Z.);; 3Cardiff School of Engineering, Cardiff University, Cardiff CF24 3AA, UK; GuH5@cardiff.ac.uk

**Keywords:** high-carbon ferrochrome, Ni45, laser cladding, microstructure, wear

## Abstract

A composite coating with enhanced mechanical properties including high hardness and excellent wear resistance was produced by laser cladding of mixed Ni45 and high-carbon ferrochrome powders on an ASTM 1045 steel substrate. Different quantities, ranging from 10 to 50 wt.% of high-carbon ferrochrome powder were added to the Ni45 powder to investigate the effect of mixture content on the cladding performance. The microstructure of the coatings were examined using scanning electron microscope, and the wear resistance was compared using a wear tester apparatus among the different cases. The results showed that the microstructure of the coating with 30 wt.% high-carbon ferrochrome content was mainly fine solid solution phase. With the increase of high-carbon ferrochrome content to 40 wt.% and above, cracks appeared on the cladding surface due to a large amount of chromium carbides formed during the process. The microhardness was enhanced remarkably by laser cladding the composite coating on the 1045 substrate, with 2.4 times higher than the hardness of the substrate when 30 wt.% high-carbon ferrochrome content was added. The best wear performance was achieved when the high-carbon ferrochrome content was 30 wt.%, demonstrating the smallest surface roughness and depth of wear marks. With further increased high-carbon ferrochrome content, microcracking and delamination were observed on the worn surfaces.

## 1. Introduction

Laser cladding is considered as a strategic technique since it can yield surface coatings of good performance, and often has superior properties over other hard facing techniques in terms of pureness, homogeneity, hardness, bonding and microstructure [1,2]. Over the last few decades, laser cladding has been widely used to improve the surface properties of metallic machined parts locally. A cladding material with the desired properties is fused onto a substrate by means of a high-power laser beam. By improving the mechanical properties of a specific surface locally with a dedicated material, one can use an ordinary cheap base material for surfaces that are not exposed to high loads [3,4]. Hence, the cladding material is crucial for a high-performance laser cladded coating. Recently, researchers have developed different materials for various applications using laser cladding technique [5,6].

Ni-based alloys, owing to their superior mechanical properties including good wettability, wear resistance, and corrosion resistance, find wide applications in industries like the aerospace, marine industries, and nuclear reactor [7]. Therefore, Ni-based alloys have been widely used in various surface coating techniques, especially laser cladding. As a Ni-based alloy, Ni45 is often being used as a coating material due to its moderate strength and toughness, good physicochemical compatibility with steel, and being easy to form a good metallurgical bonding with the substrate [8,9]. However, due to the relatively low contents of boron and carbon in the alloy, the hardness and wear resistance of the cladded coating is lower than Ni60. Therefore, Ni45 alloy is not suitable for applications where high wear resistance is required [10]. On the contrary, the high-carbon ferrochrome shows both high hardness and heat resistance. Besides being abundant, ferrochrome is advantageous in its relatively low cost. However, high-carbon ferrochrome cannot be used directly as a cladding material because of its brittleness [11,12].

This study attempts to obtain a composite coating with improved mechanical properties including crack-free, high hardness and good wear resistance by combining the advantages of both high-carbon ferrochrome and Ni45. In this paper, laser cladding of mixtures of high-carbon ferrochrome and Ni45 powders in various proportions on ASTM 1045 steel will be performed. The benefits of adding high-carbon ferrochrome into Ni45 powder will be investigated regarding the surface cracking, changes on microstructure, micro-hardness and wear resistance etc. It is expected that the composite powders developed in this study will be a promising alternative which can be used as a high hardness and high wear resistance cladding material for various industrial applications.

## 2. Experimental Materials and Procedure

Laser cladding experiments were performed on an ASTM 1045 steel substrate in the dimensions of 100 mm × 100 mm × 10 mm. The materials used for the cladding process were Ni45 and high-carbon ferrochrome powders. The chemical compositions of the Ni45 and high carbon ferrochrome powders are listed in Table 1 and Table 2, respectively. Figure 1 shows the morphologies of the two powder materials taken from a scanning electron microscopic (SEM, Merlin Compact-6170 ZEISS, Jena, Germany). Different quantities, ranging from 10 to 50 wt.% of high-carbon ferrochrome powder were added to the Ni45 powder to examine the effect of different mixture contents on the cladding performance. The powder mixtures were prepared by mechanical mixing using a QM-3SP2 planetary ball miller (Nanjing University Instrument Corp., Nanjing, China). Prior to the laser cladding experiments, the mixed powders were heated to 150 °C and kept for 30 min in a drying chamber to eliminate the moisture.

A continuous wave (CW) high power fiber laser (YLS-6000, IPG Photonics, Massachusetts, USA) of 1070 nm wavelength was used in the laser cladding experiments. A laser power of 2000 W and a scanning speed of 6 mm/s were employed as the process parameters with an overlap ratio of 35% between two adjacent tracks. The laser spot size was adopted as 5 mm. The powders were delivered by a DPSF-2 type coaxial powder feeder (Aviation Industry Corporation of China, Ltd. Beijing, China) at a rate of 14 g/min assisted by 12 L/min argon gas flow. After cladding, surface cracks were revealed by dye penetration testing. The cross-sectional microstructure of the coatings was investigated using SEM and the elemental mapping distribution was performed by an energy dispersive X-ray spectroscopy (EDS, Oxford X-Max, Oxford, UK). The micro-hardness profile along the depth of the coatings was determined by a fully automated FM-ARS900 Vickers tester (Future-Tech Corp., Kanagawa, Japan) at 0.2 kg load.

A pin on disc friction and wear tester (HT-1000, Lanzhou Zhongkai, Lanzhou, China) was used to test the wear resistance under dry and sliding friction. The samples were cut and machined to a cuboid with the size of 20 mm × 20 mm × 5 mm as the disc, while a Si_3_N_4_ ball with a diameter of 5 mm was fixed as the pin. Thereafter, the dry-sliding wear test was conducted by rotating the sample at a speed of 500 rpm for 30 min with a load of 15 N. The morphologies of the wear grooves were observed using a 3D laser confocal scanning microscope (OLYMPUS LEXT OLS4000, Olympus Corp., Tokyo, Japan).

## 3. Experimental Results and Analysis

Figure 2 shows the macrographs of the laser cladded Ni45 + high-carbon ferrochrome coatings after the dye penetration tests. It can be observed that no cracks were formed when the proportion of high-carbon ferrochrome was below 30 wt.%, as shown in Figure 2a–d. With the increase of high-carbon ferrochrome content, some cracks started to appear on the surface when adding 40 wt.% high-carbon ferrochrome content shown in Figure 2e, while more cracks appeared with elongated length propagating across multiple tracks when 50 wt.% content was added, as illustrated in Figure 2f. The formation of cracks was related to the microstructure evolution during the solidification process which will be addressed in the following discussion.

Figure 3 presents the SEM images of the transverse cross-section of the laser cladded Ni45 + high-carbon ferrochrome coatings. For pure Ni45 and Ni45 + 10 wt.% high-carbon ferrochrome coatings (Figure 3a,b), the microstructure mainly consists of γ-Ni dendrites and some multiple eutectics. With the increase of high-carbon ferrochrome contents to 20 wt.% and 30 wt.%, eutectic chromium carbides were formed, as seen in Figure 3c,d. A finer microstructure was also obtained for these two cases, especially with 30 wt.% high-carbon ferrochrome added content. This could be attributed to the fact that the eutectic carbides, formed in the previous solidification stage, became nuclei which enabled further formation and growth of the grains. With further increased high-carbon ferrochrome contents to 40 wt.% and 50 wt.%, much more chromium carbides could be observed, and these carbides became lath-shape, as shown in Figure 3e,f. Figure 4 demonstrates the EDS mapping of the laser cladded Ni45 + 40 wt.% high-carbon ferrochrome coating, which exhibits that the concentrations of Cr and C were higher in the lath-shaped structures. As known, nickel alloys with chromium, boron, and silicon contents represent a type of hard facing alloys by means of the formation of nickel, chromium borides and complex carbides as dominant primary hard phases [13,14]. In this paper, high-carbon ferrochrome was added to the Ni45 alloy, and as a result more chromium carbides could form due to the high Cr content of high-carbon ferrochrome. After the laser cladding process, the carbides were usually found to be Cr_7_C_3_ and Cr_23_C_6_ [15,16,17]. Such hard phases are very brittle, making the matrix alloy less ductile and more susceptible to the high thermal stress imposed by the high thermal gradients and high solidification rates associated to the laser cladding process [18,19]. Therefore, cracks were more likely to form in the 40 wt.% and 50 wt.% high-carbon ferrochrome laser cladded coatings due to the higher contents of Cr and C, as seen in Figure 2e,f.

Figure 5 demonstrates the microhardness profile along the depth of the laser cladded coatings. As can be noticed, the distribution of the microhardness presents differently in three main zones: the cladding zone, heat-affected zone and the substrate. The results indicate that the addition of high-carbon ferrochrome contributes to an increased hardness for all the cladding zones. The average microhardness values for 0–50 wt.% high-carbon ferrochrome coatings were enhanced by approximately 2.04, 2.30, 2.40, 2.62, 2.64, and 2.68 times compared with the 1045 substrate, respectively. In addition, the existence of chromium carbides can also increase the microhardness of the coatings [20,21].

In order to investigate the wear mechanism, the surface morphologies of the laser cladded coatings after wear test were characterized by optical profiler and the results are shown in Figure 6. As can be seen from Figure 6a–d, the wear tracks of 0–30 wt.% high-carbon ferrochrome coatings show negligible accumulation of particles or debris worn off from the friction pairs at room temperature, which indicated that the cold welding effect was weak among these cases. In comparison, the 40–50 wt.% high-carbon ferrochrome coatings exhibit noticeable accumulation of debris either in or around the wear tracks revealing that the cold welding effect was dominant at elevated temperatures. Among the six cases, 30 wt.% high-carbon ferrochrome coating has the shallowest groove with a width of 582 µm lying along the wear track as compared to the other laser cladded coatings, indicating the lowest wear rate. In contrast, the width of wear track of 50 wt.% high-carbon ferrochrome coating is 1006 µm.

Figure 7 shows the SEM images of the worn surfaces of different laser cladded coatings subjected to the sliding wear test. For the laser cladded 0–30 wt.% high-carbon ferrochrome coatings (Figure 7a–d), grooves of different depths could be observed on the worn surfaces, which were formed during the wear test as the tip of the hard ball digging into the specimen surface and plowing out the materials from the groove. The morphology of the worn surface indicates that the dominant wear mechanism was adhesive wear [22]. Among these four cases, the shallowest groove was obtained when the content of high-carbon ferrochrome was the highest at 30 wt.%, as seen in Figure 7d. This could be explained by the fact that with the increase of high-carbon ferrochrome content, the hardness of the coating would also increase which resulted in better wear resistance and hence a smoother worn surface. When further increasing the high-carbon ferrochrome contents to 40 wt.% and 50 wt.%, the major wear mechanism of the worn surfaces changed from adhesive wear to debris wear [23,24], as presented in Figure 7e,f. In addition, for the case of 50 wt.% high-carbon ferrochrome content, a large amount of micro-cracking and delamination could be observed on the worn surface, with the microcracking propagating perpendicularly to the pin sliding direction, as shown in Figure 7f. This could be attributed to the fact that the significant amount of chromium carbides formed during the cladding process would act as wear resistant skeletons. It was also noticed that the hard phase could be worn out by the local microcracking (marked a red arrow in Figure 7f) and under certain circumstances pulled out from the worn surfaces. These small hard debris could be distributed between the two sliding contact area [25,26]. Consequently, the matrix would be cut by these hard debris, resulting in an increased wear rate along with the increased sliding distance.

## 4. Conclusions

The Ni45 + high-carbon ferrochrome composite coatings were produced by laser cladding on an ASTM 1045 steel substrate. Different quantities, ranging from 10 to 50 wt.% of high-carbon ferrochrome powder were added to the Ni45 powder to examine the effect of mixture content on the cladding performance. This work has been undertaken to advance the understanding of the structure formation and properties characterization of this new composite coating. From the analysis of the experimental results, the following conclusions can be drawn:(1)Cracks appeared on the cladding surface when the added high-carbon ferrochrome content was higher than 40 wt.%. This could be related to the formation of hard phases including chromium carbides with increased C and Cr contents in the mixed powder.(2)The microhardness was enhanced remarkably by laser cladding Ni45 + high-carbon ferrochrome composite on the 1045 substrate, due to the existence of chromium carbides. It was found that with 30 wt.% high-carbon ferrochrome content, the microhardness of the composite coating was 2.4 times higher than that of the 1045 substrate.(3)The best wear performance was achieved when the high-carbon ferrochrome content was 30 wt.%, with the shallowest groove obtained lying along the wear track indicating the lowest wear rate. When further increasing the high-carbon ferrochrome content, microcracking and delamination were observed on the worn surfaces.


## Figures and Tables

**Figure 1 materials-13-01611-f001:**
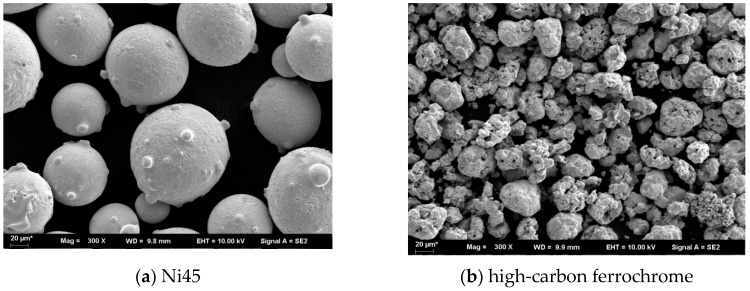
SEM Morphologies of the powders used for laser cladding.

**Figure 2 materials-13-01611-f002:**
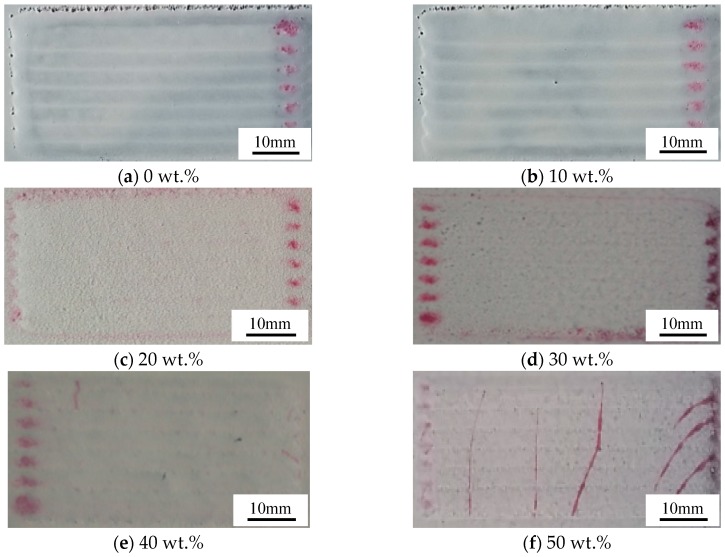
Non-destructive test results of the laser cladded Ni45 + high-carbon ferrochrome coatings.

**Figure 3 materials-13-01611-f003:**
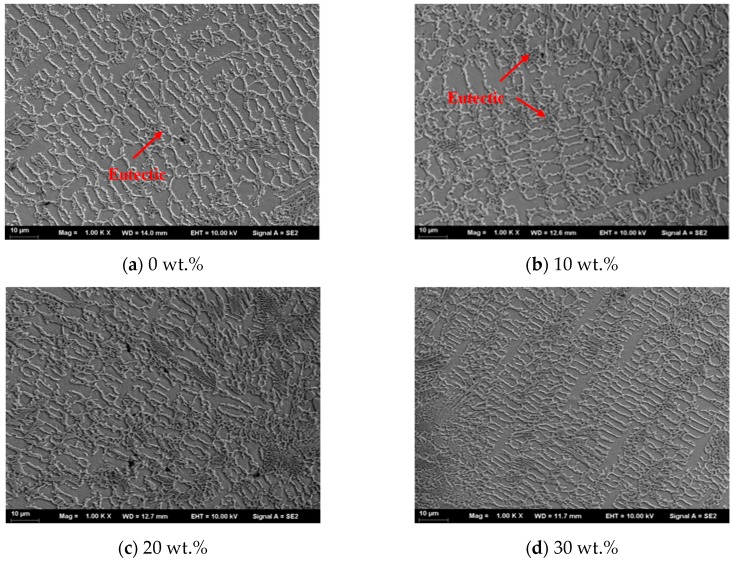
Microstructure of the laser cladded Ni45 + high-carbon ferrochrome coatings.

**Figure 4 materials-13-01611-f004:**
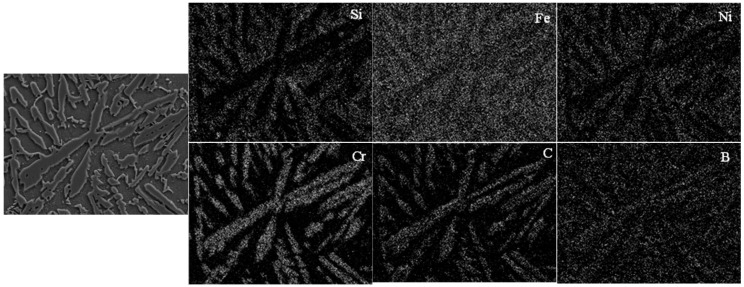
EDS mapping of laser cladded Ni45 + 40 wt.% high-carbon ferrochrome coating.

**Figure 5 materials-13-01611-f005:**
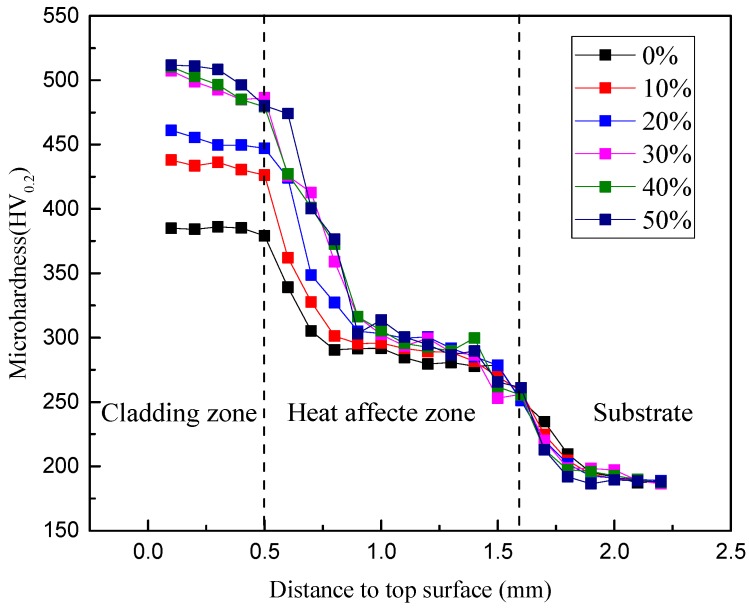
Microhardness distribution in laser cladded Ni45 + high-carbon ferrochrome coatings.

**Figure 6 materials-13-01611-f006:**
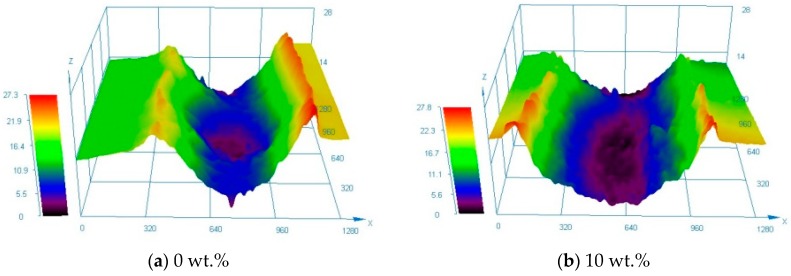
Surface morphologies of laser cladded Ni45 + high-carbon ferrochrome coatings after wear test.

**Figure 7 materials-13-01611-f007:**
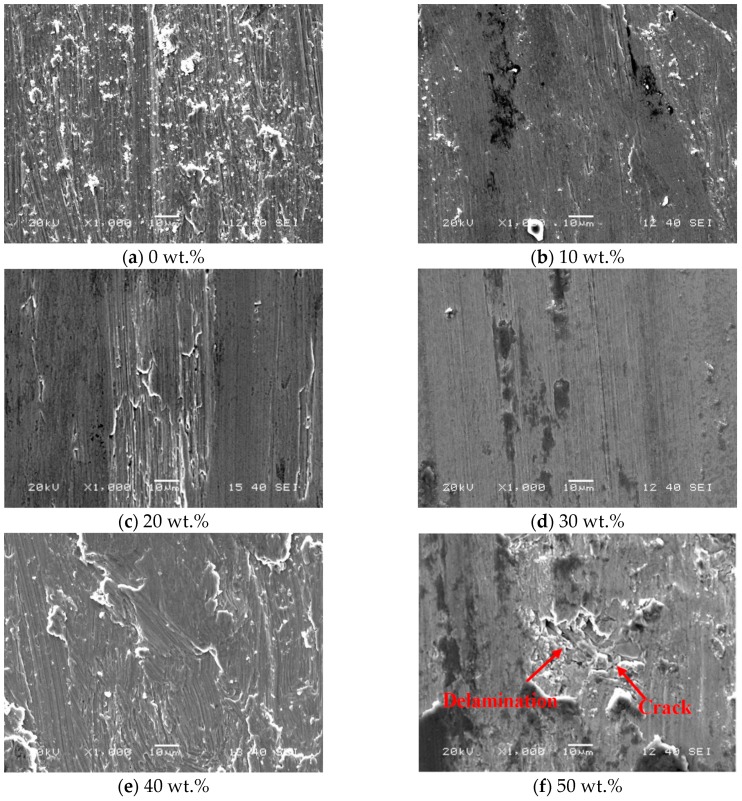
Microscopic morphologies of worn surfaces of laser cladded Ni45 + high-carbon ferrochrome coatings.

**Table 1 materials-13-01611-t001:** Chemical compositions of Ni45 nickel-based alloy powder (wt.%).

C	Cr	Si	Mn	Fe	B	Ni
0.45	12.00	4.00	0.10	10.00	2.40	Balance

**Table 2 materials-13-01611-t002:** Chemical compositions of high carbon ferrochrome powder (wt.%).

C	Cr	Si	P	S	Fe
≤6.0	≥52.0	≤3.0	≤0.04	≤0.04	Balance

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
