# Peer review of "Microstructure and Wear Behavior of Laser Cladded Ni45 + High-Carbon Ferrochrome Composite Coatings"

_materials, 2020, doi:10.3390/ma13071611_

Round 1
Reviewer 1 Report
The technical paper entitled: "Microstructure and wear behavior of laser cladded Ni45 + high-carbon ferrochrome composite coatings" (materials-756236) deals with the wear behaviour of a laser cladded coating composed of a mixture of nickel-based alloy (Ni45) and high-carbon ferrochrome, both of them in the form of powder. In my humble opinion and taking its quality into account, the present manuscript is acceptable with minor changes for its possible publication in the Journal of Materials.
Some suggestions that will help to improve the body of the manuscript are as follows:
1) Are the results for this composite coating reproducible in the case of other carbon steel substrates different from the one used in this research? Are these results reproducible for other type of steel substrates different from carbon steel? Please, clarify this point.
2) Does the type of laser (in this case, a high power fibre laser) have any influence on the results obtained by the authors? And what about the process parameters employed: power, scanning speed, overlap ratio, spot size, gas flow and type of assistant gas? Please, justify this point.
3) Figure 5 shows the micro-hardness distribution as a function of distance to top surface for the different mixture contents under consideration but what about the variability of these micro-hardness profiles plotted (for example, in terms of standard deviation values)? How many measurements were taken at each point from Figure 5?
4) Please, check Figure 3 (Microstructure of the laser cladded Ni45 + high-carbon ferrochrome coatings). At least, something happens in my PDF file as the SEM images are not presented in an orderly manner.
Author Response
1) Are the results for this composite coating reproducible in the case of other carbon steel substrates different from the one used in this research? Are these results reproducible for other types of steel substrates different from carbon steel? Please, clarify this point.
RE: The change of substrate chemical compositions will vary with the chemical compositions of the coatings due to the dilution effect during the laser cladding process. Thereafter, the microstructure and properties of the coatings will be different for a different substrate. So, a 100% reproducible coating cannot be obtained. However, the dilution for the laser cladded coating is relatively small; the results given in this manuscript can present important references for relevant studies.
2) Does the type of laser (in this case, a high power fibre laser) have any influence on the results obtained by the authors? And what about the process parameters employed: power, scanning speed, overlap ratio, spot size, gas flow and type of assistant gas? Please, justify this point.
RE: Yes, the type of laser has some influence on the experimental results. Because, the type of laser can change the laser energy density distributions, laser absorption coefficient, etc. Consequently, the macro-formation, the dilution, the microstructure and properties of the coatings may change. The process parameters employed in this study is obtained by a systematically experimental study. Experiments with different laser powers, scanning speeds and overlap ratios were conducted to obtain good macro-formation and low dilution. Some other parameters are used based on our practical experience.
3) Figure 5 shows the micro-hardness distribution as a function of distance to top surface for the different mixture contents under consideration but what about the variability of these micro-hardness profiles plotted (for example, in terms of standard deviation values)? How many measurements were taken at each point from Figure 5?
RE: In this manuscript, 3 measurements were taken at each point and the average value was given in Figure 5.
4) Please, check Figure 3 (Microstructure of the laser cladded Ni45 + high-carbon ferrochrome coatings). At least, something happens in my PDF file as the SEM images are not presented in an orderly manner.
RE: Thank you. I have changed the figure format in the revised manuscript.
Reviewer 2 Report
Paper deals with characterisation of laser cladded surface. Study is interesting , but some issues need further explanations.
Major issues:
- Hardness increase is due to precipitation of carbides and not due to grain size effect. With similar cooling conditions, coating grain size should be similar as there is no strong pinning particle in given powder mixture.
- Wear tests discuss welding effects at elevated temperatures. Were conditions for 40 and 50 wt % mixture done at different conditions? Also please try to add average area of groove for each of tested samples.
- Please discuss obtained results in light of similar studies and compare wear mechanism to other similar materials.
Minor issues:
- Please use full name for laser manufacturer, e.g. IPG Photonics.
- Fig. 3 needs to be checked as arrows are out of place.
- Mixture of Ni45 and ferrochrome should be used consistently as weight %.
Author Response
Major issues:
- Hardness increase is due to precipitation of carbides and not due to grain size effect. With similar cooling conditions, coating grain size should be similar as there is no strong pinning particle in given powder mixture.
RE: Thank you for your advice. I have deleted this description in the revised manuscript.
- Wear tests discuss welding effects at elevated temperatures. Were conditions for 40 and 50 wt % mixture done at different conditions? Also please try to add average area of groove for each of tested samples.
RE: Wear test conditions for 40 and 50 wt % mixture are the same. The elevated temperature is referring to that the temperature was high during the wear test process. Because the groove area of the test sample is very irregular, there will be a big error in the measured area. So the measured widths are added in the revised manuscript.
- Please discuss obtained results in light of similar studies and compare wear mechanism to other similar materials.
RE: It is difficult for us to find the same parameters used in the wear tests. So it is not convincing to compare with other studies. So in the manuscript, the wear mechanism of coatings with and without high-carbon ferrochrome was compared. In addition, 3 references about wear mechanisms study have been added in the revised manuscript.
Minor issues:
- Please use full name for laser manufacturer, e.g. IPG Photonics.
RE: Thank you for your advice. Photonics has been added in the manuscript.
- Fig. 3 needs to be checked as arrows are out of place.
RE: Thank you for your advice. Photos have been revised.
- Mixture of Ni45 and ferrochrome should be used consistently as weight %.
RE: Thank you for your advice. Mixture of Ni45 and ferrochrome has been used consistently as weight %.
Reviewer 3 Report
Good morning,
here are my suggestions and comments:
1) row 65: there is a write error;
2) row 97: might be usefull for the reader to read that the reason why it happen will be further in the text (row 126);
3) row 152: there isn't the description of the used Optical Profiler in experimental materials and procedure;
4) row 172: "The morphology of the worn surface indicates that the dominant wear mechanism was adhesive wear". Also in literature are there references for this result? If yes, please insert reference.
5) row 173: "Among these three cases". Perhaps I'm wrong, but are they 4?
6) row 177: "When further increasing the high-carbon ferrochrome....changed from adhesive wear to debris wear". The same of row 172. In literature, are there references for this result? If yes, please insert reference
Author Response
1) row 65: there is a write error;
RE: Thank you for your advice. The error has been deleted.
2) row 97: might be usefull for the reader to read that the reason why it happen will be further in the text (row 126);
RE: Thank you for your advice. The sentence has been added in row 97.
3) row 152: there isn't the description of the used Optical Profiler in experimental materials and procedure;
RE: Thank you for your advice. The description of the Optical Profiler has been added in experimental materials and procedure part.
4) row 172: "The morphology of the worn surface indicates that the dominant wear mechanism was adhesive wear". Also in literature are there references for this result? If yes, please insert reference.
RE: Thank you for your advice. A reference has been added.
5) row 173: "Among these three cases". Perhaps I'm wrong, but are they 4?
RE: Thank you for your advice. Yes, there are 4 cases.
6) row 177: "When further increasing the high-carbon ferrochrome....changed from adhesive wear to debris wear". The same of row 172. In literature, are there references for this result? If yes, please insert reference
RE: Thank you for your advice. 2 references have been added.